# Fungal Density in Lobomycosis in French Guiana: A Proposal for a New Clinico-Histological and Therapeutic Classification

**DOI:** 10.3390/jof9101005

**Published:** 2023-10-12

**Authors:** Geoffrey Grotta, Pierre Couppie, Magalie Demar, Kinan Drak Alsibai, Romain Blaizot

**Affiliations:** 1Dermatology Department, Cayenne Hospital, Cayenne 97306, French Guiana; geoffrey.grotta@ch-cayenne.fr (G.G.); pierre.couppie@ch-cayenne.fr (P.C.); 2UMR TBIP Tropical Biomes and Immunophysiopathology, University of French Guiana, Cayenne 97300, French Guiana; magalie.demar@ch-cayenne.fr; 3Parasitology Department, Cayenne Hospital, Cayenne 97306, French Guiana; 4Histopathology Department, Centre Hospitalier de Cayenne, Cayenne 97306, French Guiana; mohamed.drakalsibai@ch-cayenne.fr

**Keywords:** lobomycosis, Histopathology, Therapeutics, fungal disease, Neglected Tropical Disease

## Abstract

Background: Lobomycosis is a rare cutaneous tropical neglected disease caused by the fungal agent *Lacazia loboi*, recently renamed *Paracoccidioides lobogeorgii*. Our objectives were to present all cases of lobomycosis diagnosed in French Guiana, to offer a precise description of their histopathological features and to propose a new clinico-histological prognostic classification. Methods: All cases of lobomycosis diagnosed in French Guiana between 1959 and 2022 were included. We looked for associations between the occurrence of relapses and the clinic-histological form. Results: 31 patients diagnosed with lobomycosis were included. An epidemiological shift was observed in the 2000s as Brazilian patients became the most important ethnic group. Gold mining, forestry and fishing/sailing were reported as professional exposures. New histological features, such as inflammatory “rosettes” formations were described. We describe two clinic-histological patterns: a major form (high fungal density and/or multifocal lesions) and a minor form (low fungal density, unifocal lesions, association with fewer relapses). Conclusions: The changing epidemiology of lobomycosis in French Guiana is characterized by a shift towards Brazilian patients, mostly gold miners. Minor forms should be treated with surgery, major forms with a combination of surgery followed by nultiple drug therapy (MDT) or posaconazole.

## 1. Introduction

Lobomycosis is a cutaneous and subcutaneous infectious disease caused by the fungal agent *Lacazia loboi*, which has recently been renamed Paracoccidioides lobogeorgii [1]. It is responsible for a keloid-like nodular lesion usually involving the limbs and the head [2]. Its natural history is characterized by an increase in the number and size of lesions, sometimes leading to their dissemination over several body parts [3]. The number of reported cases is increasing, currently reaching 907 worldwide cases, most of them (496, 55%) in the state of Acre in Brazil and 31 (3.4%) in French Guiana [4,5]. However, lobomycosis is still considered as “the most neglected of Neglected Tropical Diseases” and suffers from a clear lack of clinical and therapeutic data [4].

The disease mostly affects men living or working in the forest areas of the Amazon, following inoculation through small skin traumas [6,7,8]. The main hypothesis is that the reservoir of the fungal agent is found in tropical waters and on trees in the Amazon rainforest and could also contaminate animals and tools. Indeed, a history of trauma is often found years before the appearance of lobomycosis lesions. It has been reported that animal bites (such as fish or insects) or cuts by plants or sharp objects would lead to inoculation [4,5,9].

The diagnosis is based on the presence of the fungal agent on direct parasitological examination or on histological examination of a skin biopsy [10,11]. To date, no spontaneous recovery has been described [4,12]. In cases of single or multiple small lesions, surgical removal is the best therapeutic alternative. Safety margins are mandatory due to the high rate of local relapse [5,13,14,15]. If surgery is impossible (for large and/or multiple lesions), systemic treatment is indicated. Antifungal agents are not very effective, with a high relapse rate, especially in disseminated forms [7,16,17,18,19]. After a long series of case reports with only limited proof, recent studies have shown the efficacy of leprosy MDT in larger cohorts [18,20].

French Guiana is a French overseas territory in South America. It is mostly covered by the Amazon rainforest, an environment where lobomycosis is known to thrive. However, only a few cases have been reported in this territory since the first known case in 1959 [21,22,23,24,25]. There is a clear need to collect, gather and analyze data concerning the occurrence and treatment of lobomycosis in French Guiana.

Histological examination in lobomycosis typically shows a granulomatous inflammatory infiltrate with multi-nucleated giant cells, usually associated with a high amount of intra- and extracellular yeasts [26]. *Lacazia loboi* appear as numerous double-walled spherical yeasts measuring 5 to 11 µm, isolated or distributed branched chains of 2 to 8 cells, connected by short tubules. However, there are few data on the different histological patterns of lobomycosis in skin biopsies. The evolution of histological characteristics through the different stages of the disease and its treatment has not been described. It is not known if the fungal density or other histological features can be linked to a better response to treatment or to severe forms of disease.

The main objective of this study was to describe the clinical and therapeutic features of lobomycosis in French Guiana from 1959 to 2022. Our secondary objectives were to offer a precise description of the histopathological features of lobomycosis cases in this territory and to propose a prognostic clinico-histological classification to guide therapeutic management.

## 2. Materials and Methods

### 2.1. Inclusion

All patients diagnosed with lobomycosis in French Guiana between 1959 and 2022 were included. As the Cayenne Hospital Center is the only referral center for skin diseases and works closely with all other health care providers, all cases of lobomycosis or other subcutaneous mycoses are systematically referred to its Dermatology Department. Suspicions of lobomycosis in French Guiana are always confirmed either by the Mycology-Parasitology Laboratory or the Histopathology Laboratory of the Cayenne Hospital Center. Diagnosis was made by detection of *Lacazia loboi*/*Paracoccidioides lobogeorgii* yeasts in histological examination of a skin biopsy or by direct examination of a skin smear after Gomori-Grocott staining. Data were retrospectively collected from the software and archives of the Dermatology, Histopathology and Parasitology-Mycology Departments. An exhaustive search of all coded diagnoses of lobomycosis was conducted through the three respective databases.

### 2.2. Data Collected

The following data were extracted:socio-demographic characteristics (age, gender, ethnicity, country of origin, occupation, possible exposure factors, date of arrival in French Guiana)clinical characteristics (date of lesions onset, skin trauma, duration of disease, size, number and body location of the lesions, lymphadenopathy)treatment history (abstention, local or systemic treatment, duration of treatment, drugs/treatments used)outcome. The different responses to treatment were defined according to the method in Gonçalves [20], as follows: partial response (50% decrease or atrophy of lesions at the last follow-up visit), complete response (total regression of lesions at the last follow-up visit without relapse for 2 years after the end of treatment). Treatment failure included cases with no response (<50% decrease or increase in number or size of lesions) and cases with relapse (resurgence of lesions after initial remission); data were collected on the type and number of relapses, as well as the presence of a local (on the site of the previous lesion) or distant relapse.

### 2.3. Histopathology

The formalin-fixed, paraffin-embedded (FFPE) blocks and slides containing the lobomycosis’s skin specimens were retrieved for all those available from the archive of the Pathology’s Department of Cayenne Hospital Center. Histological sections of 3–4 µm were made with a microtome HM 355S (MicromMicrotech France, Brignais, France) from the FFPE blocks, then two slides for each specimen were stained by Hematoxylin and Eosin and Gomori-Grocott stains.

Images were obtained by a slide scanner (Pannoramic 250, 3DHistech, Budapest, Hungary) and analyzed using the 3DHistech softwares V 2.1 (casecenter and caseviewer, 3DHistech, Budapest, Hungary). To look for associations between clinical and histological features, an anonymous number was assigned to each histological block and the slides were blindly analyzed by a reference histologist specialist in lobomycosis. We analyzed the type of inflammatory infiltrate on each Hematoxylin and Eosin (HE)-stained slide (multinucleated giant cells, histiocytes, lymphocytes, plasma cells, neutrophils and eosinophils) and the location of this infiltrate (dermis, hypodermis or both).

Since the objective ×40 (400× magnification) of the light microscope is the most used by pathologists in daily practice, we calculated the fungal density (Gomori-Grocott +) or number of yeasts per ×40 field. Density was defined as the amount of yeast per field at ×400 magnification over an average of five fields in the superficial and deep dermis. Hotspot fungal density corresponding to the highest fungal density field was measured in each slide. Density in hotspots was classified as low (<400 yeasts/field), medium (400–800) and high (>800/field). We also measured on each Gomori-Grocott slide the size per µm of the smallest and largest yeasts and determined the location of the yeasts (intracellular, extracellular or both), and described the predominant pattern of yeast arrangement for each specimen.

### 2.4. Ethics

All patients gave their written consent for research purposes on their skin biopsies. All patients were also reached out to by phone or mail to collect their consent regarding the use of their clinical data. Informed consent was obtained from all subjects involved in the study. According to French law, no further legal clearance was needed. This study was declared on the Health Data Hub with the accession number F20220817203802.

### 2.5. Statistical Analysis

Associations were explored between the presence of a low fungal density and the risk of multifocal or unifocal presentations using Fisher’s exact test or χ^2^ test. Associations were also sought between the mean fungal density in hotspots and the number, size and presentation (unifocal or multifocal) of lesions. We also looked for associations between epidemiological features and the date of diagnosis (see below). Associations with a *p*-value < 0.05 in univariate analysis were deemed significant. Stata^®^ software Version 12.0 (Statacorps, College Station, TX, USA) was used for the statistical analyses.

## 3. Results

### 3.1. Patient’s Characteristics

A total of 31 patients were included in the study. Their characteristics are summarized in Table 1. The last patient in the 20th century was diagnosed in 1992, with no new case until 2010. Considering that this time gap might reflect a chronological shift in epidemiological trends, we decided to split and compare clinical and histological data between these two historical groups (Group 1, 1959–1992, 15 patients; and Group 2, 2010–2022, 16 patients). In the total population of 31 patients, the overall median age at diagnosis was 55 years (45–62 yrs). There was a male predominance (26/31, 84%).

When comparing the two historical groups, the most frequent ethnic origins in Group 1 were Creole (8/15, 53.3%) and Maroon (5/15, 33.3%), while Group 2 was clearly dominated by Brazilian patients (13/16, 81.3%). There was no Brazilian patient in Group 1.

Among the 28 patients whose occupation was known, twenty-one (75.0%) had significant occupational exposure to the rainforest or aquatic Amazon environments, such as gold panning (12/28, 9%), fishing/sailing (6/28, 21.4%) and forestry/logging (3/28, 10.7%).

However, there was a larger proportion of gold miners in Group 2 (9/16 patients, 56.3%) compared with 3/15 (20.0%) in Group 1.

In univariate analysis, Group 1 was associated with a Creole origin (OR 17.14, [95% CI 1.59–809.71]; *p* = 0.0039), while the occurrence of Brazilian patients (OR 0.02, 95% CI [0.01–0.21]; *p* ≤ 0.0001) and gold mining (OR 0.19, 95% CI [0.03–1.19]; *p* = 0.0384) as an occupation were significantly rarer than in Group 2.

### 3.2. Clinical Features at Diagnosis

A factor of exposure by skin trauma (wound, cut, sting or bite) prior to lesion onset was found in 23% of patients (7/31). The average duration of symptoms before diagnosis was 12 years for Group 1 versus 4 years for Group 2, with an overall average of 7 years (0.5–40 yrs). The clinical characteristics between the two groups were broadly similar. At the time of diagnosis, a single lesion was observed in 42% of cases (13 patients).

The lesions were multiple but unifocal (several lesions located on a single body part) in 22.5% of cases and multifocal (several lesions involving at least two different body parts [18]) in 35.5% of cases. The most frequent locations of lesions were the lower limbs (55%), followed by the upper limbs (45%) and the head (23%). Of note, among the seven patients with head involvement, almost all lesions (six patients, 85.7%) involved the right ear.

Most patients had unilobed or multilobed nodular lesions (90%, 28/31), two patients had ulcerated lesions, and one patient had an atypical lesion on the cheek in the form of a centimetric hyperpigmented macule (Figure 1B). Clinical pictures are gathered in Figure 1.

Lesion size ranged from 1 to 13 cm, with a mean of 4.6 cm and a median of 3.5 cm.

The lesion could be hyperpigmented (10%), hypopigmented (10%), depigmented (10%), or pinkish (13%); in the other cases, no information on pigmentation was specified.

Seven patients (22.6%) had lymphadenopathy, including three patients with multiple lymph node locations. A lymph node sample was collected for three patients and showed yeasts on microscopic examination in two cases. Only one patient presented with an altered general condition.

Blood samples were taken for eleven patients and were always normal, except for two patients with hypereosinophilia.

### 3.3. Evolution

The mean follow-up time was 5.8 years, with a median of 17 years. Eighteen patients (58.1%) received local or systemic treatment as first-line therapy, six patients (19.4%) were lost to follow-up before any treatment, and seven patients (22.6%) received no treatment as therapeutic abstention was chosen (six in Group 1 and one in Group 2).

Among the abstentions, no spontaneous remission was observed; one patient was lost to follow-up, and all the others progressed.

Six patients out of 31 (19.4%) received local treatment without systemic antifungals, five of them by surgical excision. All of them relapsed with a mean duration of 7 years (0.80–12.33).

Twelve patients (38.7%) received systemic treatment at the first line. The results are detailed in Table 2. The only complete response was observed in a patient with a mild form, treated with surgery associated with oral terbinafine. There was no relapse after 12 years of follow-up. No deaths attributable to lobomycosis were observed.

### 3.4. Histological Analysis

A total of 21 samples (blocks and slides) of lobomycosis were available in the archives of the Pathology Department, corresponding to 13 different patients. Eight histological blocks corresponded to follow-up samples. Histological patterns and fungal densities were analyzed in the 21 available samples. Analysis of the skin tissue at ×400 magnification revealed a dense granulomatous inflammatory infiltrate with lymphocytes, neutrophils, plasma cells, and multinucleated giant cells of the resorptive type in 19/21 (90.5%) cases. Two of 21 (9.5%) had an inflammatory infiltrate composed of numerous neutrophils and eosinophils and lympho-histiocytes of the telangiectatic scar type.

The lesion was located in both the dermis and hypodermis in 13 cases (62%) and in the dermis only in 8 cases (38%). The average minimum diameter of the yeasts was 5.1 µm (3–6.7 µm), and the average maximum size was 11.75 µm (7.8–12.4 µm). 

The yeasts were both extracellular and intracellular, except for two cases of telangiectatic cicatricial type, where they were only extracellular (Figure 2).

We found three patterns of yeast arrangement by Gomori-Grocott stain; a small chain pattern of <5 yeasts in 10/21 (48%) samples, corresponding to 13 patients (%) and 48% of cases: a long chain pattern of ≥5 yeasts for four cases (19%); and a rosette pattern for seven cases (33%), corresponding to a round chain arrangement leaving a space in the center of the structure that usually has >5 yeasts. The different patterns of lobomycosis yeast arrangements are presented in Figure 3.

As fungal density did not vary between initial and follow-up samples, we used the initial fungal density for statistical analysis (13 samples). The mean fungal density was 464 (80–1074) yeasts per field in the superficial dermis, 321 (50–716) in the deep dermis, and 628 (128–1639) in hotspots. Respectively, the median fungal density was 436 (351–559), 277 (102–450), and 562 (288–758) yeasts per field.

All patients with low fungal density (Hotspot < 400 yeasts per field) had unifocal involvement. Five patients had a low fungal density; all of them had unifocal presentation. Among eight patients with a medium or high density, five had a plurifocal presentations and three had a unifocal one. A low fungal density was associated with a lower risk of multifocal involvement (OR 0.1, 95% CI, [0.01–0.68]; *p* = 0.044). The only complete response of the cohort was observed in one of these patients (Figure 1A).

Concerning patients with high fungal density (Hotspot > 800 yeasts per field), all of them had multifocal involvement.

Concerning the patients with a medium density (Hotspot 400–800 yeasts/field), the number and size of lesions seemed to increase concomitantly with fungal density. However, these associations were not statistically significant. Concerning patients with a partial response, no variation in fungal density was observed throughout the duration of treatment.

## 4. Discussion

### 4.1. Epidemiologic Data

One of the strengths of this study is its long-term range, with inclusions spanning more than half a century. This allowed us to observe a surprising gap without any new cases for 20 years. The incidence of lobomycosis probably decreased at the end of the 20th century as living conditions in French Guiana improved and gold mining declined. However, in the 2000s, a spectacular rise in the price of gold led to a resurgence of gold mining in the French Guiana hinterland. This led to a second wave of massive Brazilian immigration motivated by illegal gold panning. This could explain why most cases in Group 2 were observed in people of Brazilian origin (80%). On the other hand, most patients in Group 1 were Creole or Surinamese. In addition, more than 90% of patients in both groups had professional exposure involving the Amazon environment, notably aquatic and forestry activities. This supports the hypothesis that the transmission mode of *Lacazia loboi* would be linked to traumatic contact with trees or other aquatic material contaminated by the yeasts [4,18,27]. It is not known if these Brazilian patients acquired lobomycosis in French Guiana or were infected in Brazil prior to their arrival in French territory. A phylogenetic analysis comparing the strains isolated in French Guiana with those identified in Brazil could help answer this question [28,29].

### 4.2. Clinical Data

The clinical history of the disease is characterized by a slow evolution in size and number of lesions. Different classifications have been proposed for lobomycosis: some authors differentiate single, multiple or disseminated forms according to the number of lesions [18,20], while others prefer to classify according to the unifocal or multifocal character [11,27].

However, while some patients will show only local progression, others will progress from a mild form to a disseminated one [6,19]. There is currently a lack of prognostic data on which patients are likely to present disseminated lesions at a later stage, which is an important factor for disease severity. Indeed, though lobomycosis is a benign pathology, the evolution of some patients towards disseminated forms is responsible for major functional and aesthetic impairment [3,19,25].

The interest of this new classification is to propose an optimal therapeutic strategy for patients at risk of later dissemination. This prognostic tool will be particularly useful in remote settings and low-resource populations, where compliance and follow-up are often poor.

In our study, most patients with multifocal involvement at diagnosis presented a resistant form with either no response to the different therapeutic lines or an important relapse after treatment, as previously reported [18,20]. Conversely, it was noted that patients with unifocal involvement had a better response to treatment or presented only local relapses.

In our study, nearly a quarter of patients had satellite lymphadenopathy. Lymphatic involvement with histological evidence of yeasts in lymph nodes has already been reported in the literature [6,30,31]. In view of these data, it would seem more concrete to differentiate unifocal forms from the multifocal forms disseminated to multiple parts of the body, which could reflect a process of lymphatic dissemination.

### 4.3. Pathological Data

To date, a correlation between inflammatory patterns in histology and clinical presentations has been shown in only one study [26]. In this work, we report a specific inflammatory pattern with yeasts presenting in an original “rosette” formation, which concerned 33.3% (7/21) of patients. This pattern has already been described in other invasive fungal infections such as histoplasmosis [32], and reflects an inflammatory process leading to an explosion of extra-cellular yeasts.

We present for the first time a precise observation of fungal density in lobomycosis, with great variability between patients. A low hotspot fungal density was associated with a lower risk of multifocal dissemination, which is in line with our hypothesis that fungal density could be used as a predictive factor to adapt therapeutic strategies.

Although there was no significant association with disease progression, probably due to the common lack of available data, there were several interesting findings regarding the evolution of the patients and their response to treatment:Patients with low fungal density (<400 yeasts/field) had single or multiple localized involvements. They all showed a partial or complete therapeutic response, and none progressed to a major form.Patients with high fungal density (>800 yeasts/field) presented a multifocal form and did not respond to first-line treatment.Concerning patients with medium fungal density (400–800 yeasts/field), they had unifocal or multifocal forms, and those who received treatment had a partial response. Some of them were still under treatment at the time of the study, and adjuvant surgery could be performed after debulking with antifungals.

The bacillary index is the cornerstone of therapeutic guidelines in leprosy, as treatment duration is different between paucibacillary and multibacillary patients [33]. Likewise, we propose to classify lobomycosis patients into two categories according to their fungal density and clinical status: a minor form corresponding to patients with unifocal lesions associated with a low fungal density, and a major form corresponding to multifocal forms and/or with a medium to high fungal density.

### 4.4. Therapeutic Strategy

In view of these prognostic data, we propose a new therapeutic algorithm for lobomycosis in Figure 4. Considering the risk of progression after an initially mild presentation, we think that abstention should not be recommended in lobomycosis.

For minor forms, we propose surgical management with safety margins. Patients with major forms should be treated with a combination of excisional surgery and adjuvant systemic treatment. Systemic treatment should be based on MDT, or posaconazole which has shown efficiency in several case reports [17,34].

For lesions that are not accessible to surgery (large lesions or involving the face), a debulking systemic treatment with MDT or posaconazole will be proposed, followed by excisional surgery after a decrease in size (Figure 5).

Diagnosis of lobomycosis by direct examination of skin smears (Figure 6) enables rapid management of patients, with treatment instituted within a few days of diagnosis. Although histological examination on skin biopsy takes longer to obtain, initial management can be corrected secondarily according to the classification: proposal of adjuvant treatment in the case of a single lesion that has been operated and showing high fungal density and a high risk of recurrence (major form).

One of the strengths of this study is the depth of the histological research work and its original combination with clinical data. This study included patients over more than half a century, which gave us a large overview of the epidemiological shifts concerning lobomycosis in French Guiana. One of the major biases was the high number of patients lost to follow-up due to limited health care access in the remote areas of French Guiana. As the Cayenne Hospital is the only referral center for dermatology, all cases of lobomycosis diagnosed during the study period were necessarily included. However, it is possible, due to barriers to health care access, that patients were not diagnosed during the study period. However, this limitation is often present in lobomycosis [4,20]. Another limitation is that immunohistochemistry was not available for this study. Fungal density and histopathological patterns may be related to immune responses. Further studies should look for the role of host factors through measures of antibodies and cellular immunity.

## 5. Conclusions

This study shows how the changing epidemiology of lobomycosis in French Guiana was characterized by a shift towards Brazilian patients, mostly gold miners. We also use a combination of clinical and histological patterns to propose a new classification of lobomycosis cases, based on prognostic factors and with therapeutic implications.

## Figures and Tables

**Figure 1 jof-09-01005-f001:**
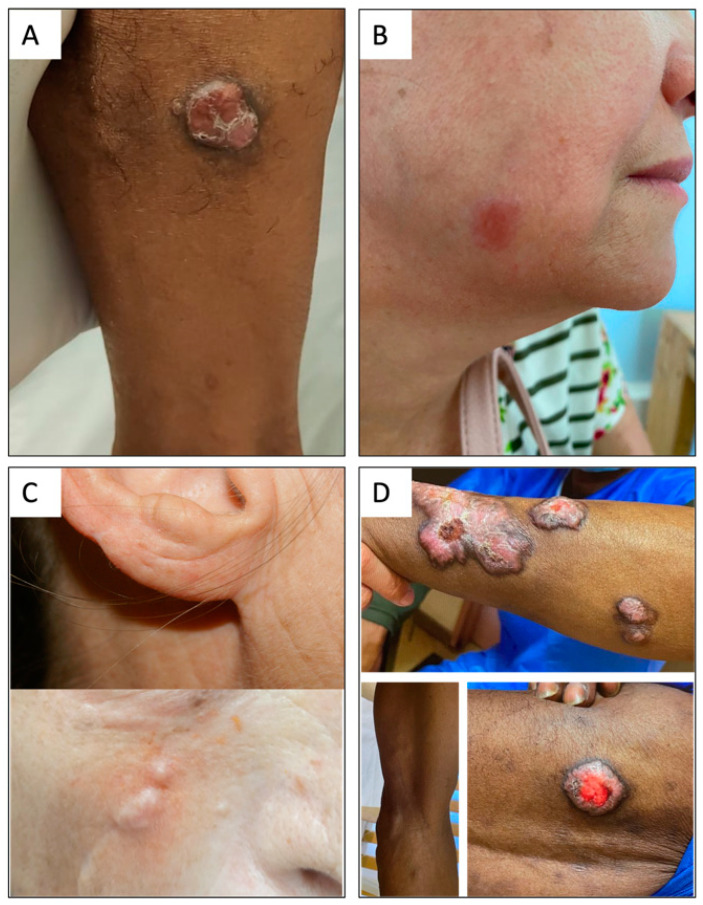
Clinical aspects of lobomycosis cases in French Guiana. (**A**) Minor form: unifocal lesion with low fungal density and a complete response. (**B**) Major form: unifocal but medium-density, clinical relapse. (**C**) Major form: small but multifocal lesions and high fungal density; clinical relapse after multiple surgeries. (**D**) Typical major form: multifocal severe and high density, clinical relapse.

**Figure 2 jof-09-01005-f002:**
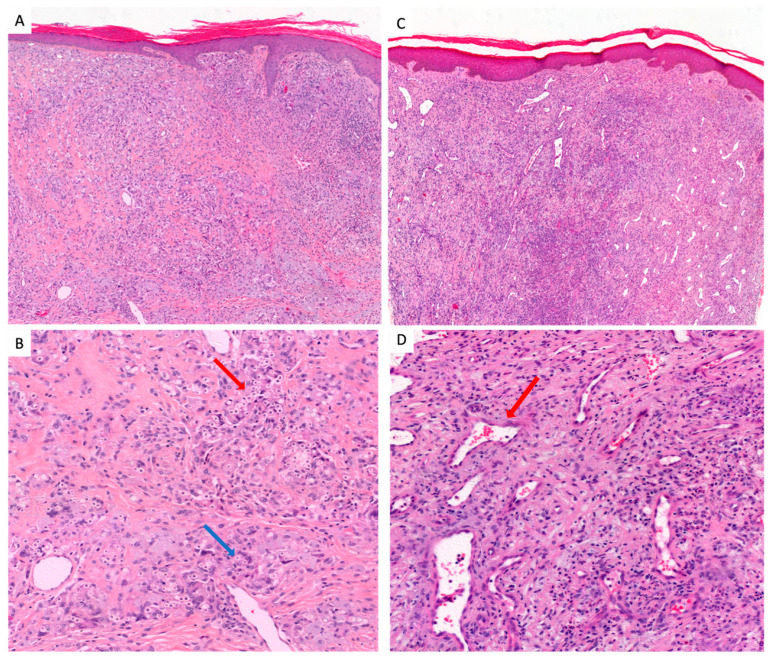
The two histological types of lobomycosis. (**A**) Classic type: acanthosis and hyperkeratotic epidermidis, important inflammatory reactions. A milder inflammatory infiltrate is also seen in the center and periphery of the lesion, consisting of a few lymphocytes, plasma cells and rare eosinophils (HE-stain 10X). (**B**) Classic type: reaction of the dermis composed of multinucleated giant cells (blue arrow) and histiocytes with numerous round yeasts (red arrow), mostly intracellular with some extracellular yeasts (HE stain 40X). (**C**) Telangectatic scar type: the dermis is particularly fibrous and looks like a scar that delimits numerous dilated vessels (red arrow). The scar is punctuated by a moderate inflammatory infiltrate (HE-stain 10X). (**D**) Telangectatic scar type: a moderate inflammatory infiltrate consisting of histiocytes, lymphocytes and plasma cells with exceptional multinucleated giant cells with rare yeasts mostly intracellular but numerous dilated vessels showed by the red arrow (HE stain 40X).

**Figure 3 jof-09-01005-f003:**
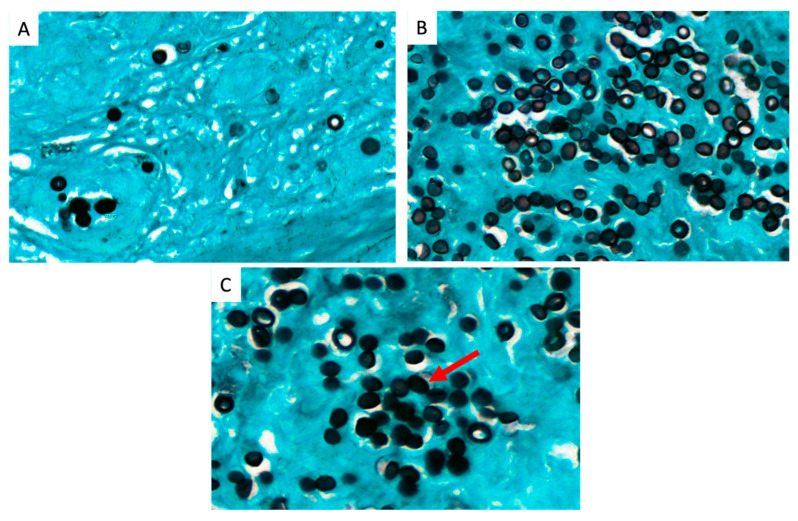
Three patterns of lobomycosis yeast arrangement. (**A**) small chain pattern of <5 yeasts (Gomori-Grocott stain 40X). (**B**) long chain pattern of >5 yeasts (Gomori-Grocott stain 40X). (**C**) rosette pattern corresponding to a round structure that usually has >5 yeasts shown by the red arrow, Gomori-Grocott stain 60X).

**Figure 4 jof-09-01005-f004:**
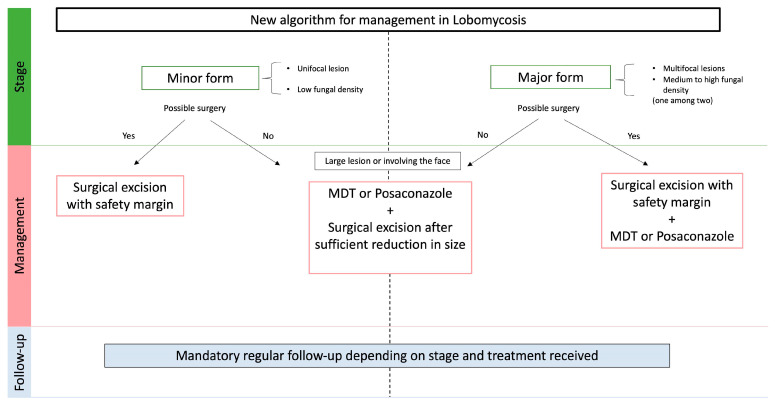
Algorithm for the management of lobomycosis, based on clinic-histological classification.

**Figure 5 jof-09-01005-f005:**
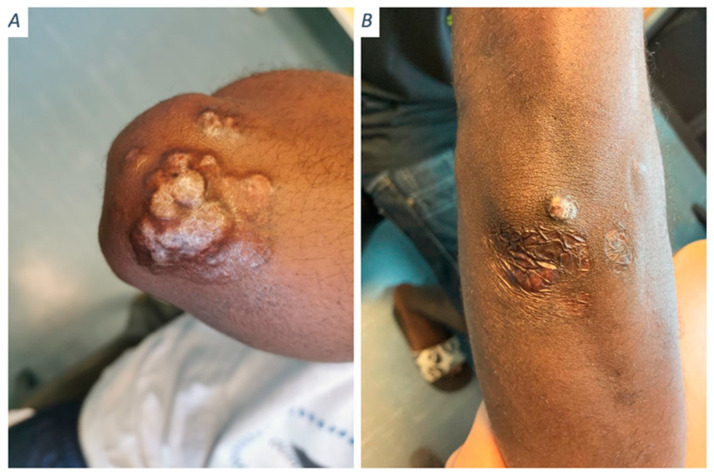
Management of patients with major forms of lobomycosis. (**A**) Major form with multiple unifocal lesions and medium fungal density. (**B**) Partial response after 18 months of daily 800 mg of posaconazole. The patient is still under treatment and will undergo adjuvant surgery on the last persisting node.

**Figure 6 jof-09-01005-f006:**
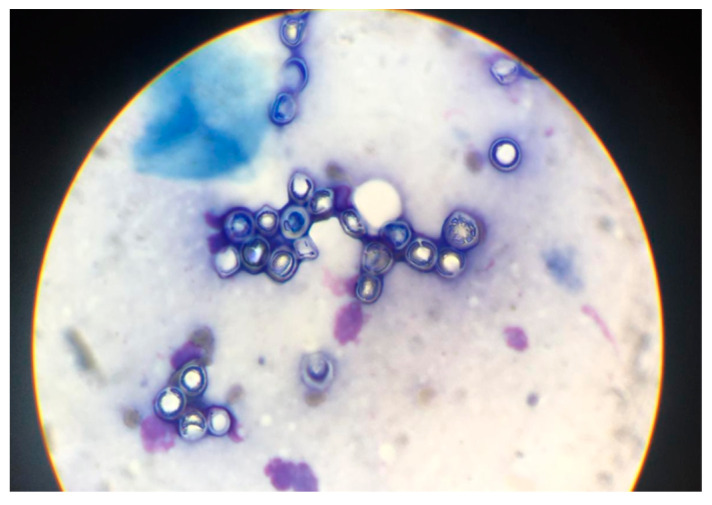
Diagnosis by direct examination of skin smear: rounded yeasts about 5 to 8 μm long with a fungal double wall, arranged singly or in chains with inter-yeast bridges in microscopy.

**Table 1 jof-09-01005-t001:** General characteristics of lobomycosis cases in French Guiana, 1959–2022 (n = 31), and comparison of epidemiological features between the two historical groups.

	Group 1	Group 2	Total	OR [95% CI]	*p*-Value
Year of Diagnosis	1959–1992 n = 15	2010–2022 n = 16	Total n = 31		
**Age**					
Mean (min–max)	61 (45–83)	49 (33–75)	54 (33–83)		
Median (Q1–Q3)	60 (53–65)	46.5 (40.5–57)	55 (43–62)		
<60 years	7	15		17.14 [1.59–809.71]	**0.0039 ***
≥60 years	8	1			
**Gender (%)**					
Women	1 (7.0)	4 (25.0)	5 (16.1)	4.67 [0.37–245.87]	0.1655
Men	14 (93.0)	12 (75.0)	26 (83.9)		
**Country of birth (%)**					
Brazil	1 (6.7)	13 (81.3)	14 (45.2)	0.02 [0.01–0.21] ^1^	**<0.0001 ***
French Guiana	7 (46.7)	1 (6.3)	8 (25.8)		
Suriname	4 (26.7)	1 (6.3)	5 (16.1)		
France (mainland)	0 (0)	1 (6.3)	1 ((3.2)		
Guyana	1 (6.7)	0 (0)	1 (3.2)		
India	1 (6.7)	0 (0)	1 (3.2)		
Saint Lucia	1 (6.7)	0 (0)	1 (3.2)		
**Ethnicity (%)**					
Brazilian	0 (0)	13 (81.3)	13 (41.9)	17.14 [1.59–809.71] ^2^	**0.0039 ***
Creole	8 (53.3)	1 (6.3)	9 (29.0)		
Maroon	5 (33.3)	1 (6.3)	6 (19.4)		
Amerindian	1 (6.7)	0 (0)	1 (3.2)		
Caucasian	0 (0)	1 (6.3)	1 (3.2)		
Unknown	1 (6.7)	0 (0)	1 (3.2)		
**Occupation (%)**					
Gold digger	3 (23.1)	9 (60.0)	12 (42.9)	0.19 [0.03–1.19] ^3^	**0.0384 ***
Fisherman/sailor	4 (30.7)	2 (13.3)	6 (21.4)		
Farmer	2 (15.4)	2 (13.3)	4 (14.3)		
Logging/forestry	3 (23.1)	0 (0)	3 (10.7)		
Other	1 (7.7)	2 (13.3)	3 (10.7)		
**Duration before diagnosis** **(years)**					
Mean (min–max)	12 (0.5–40)	4.25 (0.23–20)	7.24 (0.5–40)		
Median (Q1–Q3)	8.25 (3–10)	2.33 (1–4)	3.5 (1.5–9)	4.67 [0.47–60.97]	0.1054
≥10 months	4	2			
<10 months	6	14			
**History of skin trauma n** **(%)**					
	3 (20)	4 (25)	7 (22.6)		
**Number of lesions n (%)**					
Unique	6 (40)	7 (43.8)	13 (41.9)		
Multiple unifocal	5 (33.3)	2 (12.5)	7 (22.6)		
Multifocal	4 (26.7)	7 (43.8)	11 (35.5)		
**Topography n (%)**					
Head	3 (20.0)	4 (25.0)	7 (22.6)		
Lower limbs	9 (60.0)	8 (50.0)	17 (54.8)		
Upper limbs	6 (40.0)	8 (50.0)	14 (45.2)		
Trunk	2 (13.3)	2 (12.5)	4 (12.9)		
**Type of lesion n (%)**					
**Nodule**	13 (86.7)	14 (87.5)	27 (87.1)		
**Plaque**	0 (0)	1 (6.25)	1 (3.2)		
**Macule**	0 (0)	1 (6.25)	1 (3.2)		
**Ulcer**	2 (13.3)	0 (0)	2 (6.5)		
**Adenopathy n (%)**	4 (26.7)	3 (18.8)	7 (22.6)		

^1^: The association was explored between the country of birth (Brazil vs. all others) and the occurrence in Group 1. ^2^: The association was explored between the ethnic origin (Creole vs. all others) and the occurrence in Group 1. ^3^: The association was explored between the occupation (gold mining vs. all others) and the occurrence in Group 1. *: statistically significant with *p* < 0.05.

**Table 2 jof-09-01005-t002:** Treatment outcomes for lobomycosis in French Guiana, 1959–2022.

Treatment	Daily Dosage	Duration	Number	Therapeutic Response			Follow-Up Time after Treatment
				Complete Response	Partial Response	Treatment Failure	
5-Fluorocytosine	100–150 mg/kg	1–3 mths	3	0	0	3	12 yrs
Sulfamethoxypyridazine	500 mg	1 mth	1	0	1	0	7 yrs
Miconazole injection	0.5 mL/inj/mth	5 mths	1	0	0	1	7 mths
Terbinafine	500 mg	2–6 mths	2	0	0	2	6 mths
Terbinafine + Surgical excision	500 mg	2 mths	1	1	0	0	12 yrs
Itraconazole	200–400 mg	1–4 mths	3	0	0	3	1–4 mths
Itraconazole Clofazimine + Surgical excision	400 mg 100 mg	1 mth	1	0	0	1	3 yrs
Posaconazole	800 mg	18 mths	1 **	0	1	0	18 mths
MDT *	-	1–12 mths	3 **	0	1	2	12 mths
Total			16	1	3	12	

* Multiple drug therapy consisted of monthly supervised doses of 600 mg rifampicin, 300 mg clofazimine, and 100 mg dapsone, in addition to daily doses of 50 mg clofazimine and 100 mg dapsone. ** second-line therapy.

## Data Availability

All necessary data are available in the manuscript.

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
