# Peer review of "Fungal Density in Lobomycosis in French Guiana: A Proposal for a New Clinico-Histological and Therapeutic Classification"

_jof, 2023, doi:10.3390/jof9101005_

Round 1
Reviewer 1 Report
Comments
The authors have presented a well-written manuscript of a meticulously conducted study that evaluated the epidemiological and clinical features of 31 patients with lobomycosis in French Guiana. The histopathological examination was subjected to a reanalysis, leading the authors to identify two distinct clinical-histological patterns. In light of this, they propose a novel classification for the disease comprising two forms: the major form characterized by high fungal density and/or multifocal lesions, and the minor form characterized by low fungal density and unifocal lesions. Nonetheless, the manuscript could be further enhanced through improvements and clarifications. Hence, I would like to recommend the following adjustments:
Major review
1. The authors state that “All patients diagnosed with lobomycosis in French Guiana between 1959 and 2022 were included”. How can the authors be certain that no other cases of lobomycosis existed in French Guiana during that time? What methodology was employed for the case search? Was an exhaustive search conducted across all healthcare facilities, as well as pathology and microbiology laboratories? Addressing this point is crucial, as the absence of cases was interpreted as a lack of occurrences. Could it be plausible that restricted access to the study hospitals during this period might have influenced the apparent absence of cases?
2. According Talhari & Talhari (2012) https://doi.org/10.1016/j.clindermatol.2011.09.014 , the diagnosis of lobomycosis can be effectively conducted through a direct examination of skin smears. This method is not only more accessible than histopathological examination, but also easier and more cost-effective. The authors of the study present a novel classification system that relies on histopathological findings. However, it's worth noting that histopathological examinations require the expertise of highly skilled medical professionals. Unfortunately, access to such professionals might be limited, especially in areas where the disease is endemic. In line with the authors' proposed algorithm, the clinical management of lobomycosis depends on the fungal density as determined through histopathological examination. Could the requirement for a histopathological examination potentially result in delays in diagnosis and treatment? Please, include a discussion about this topic in the manuscript."
3. Why did the authors refrain from conducting statistical analyses to evaluate the association between the variables under study and the diagnosis period (group 1 and group 2) (results, line 143, table 1)? This would further substantiate the claim that there were a greater number of Brazilian patients and gold digger in group 2.
Minor considerations:
Abstract
Line 23. “Conclusions: A resurgence of gold mining by Brazilian migrants in the 2000s changed the epidemiology of lobomycosis in French Guiana.” This statement presents an assumption that goes beyond the results obtained, a practice that should be avoid in the conclusion. Note that the same problem is on line 342 (conclusion section)
Line 25. Please, note that the acronym MDT has not been spelled out before.
Methods
Line 107. Please, note that the acronym HE has not been spelled out before.
Line 126. What test was performed to analyses “the association between the presence of a low fungal density and the risk of multifocal or unifocal presentations”? In the “results section”, line 234 it is unclear. (95% CI, [0.1-0.86]; p = 0.0476).
Line 130. “The relationship between individual risk factors and Lobomycosis infection was analyzed using Fisher’s exact test or χ2 test.”. There is a mistake here. The analysis did not involve assessing the associations between individual risk factors and Lobomycosis infection.
Results
Lines 166 to 168 “(Fig. 1A : Minor form. Unifocal lesion with low fungal density, complete response. Fig. 1B : Major form. unifocal but medium density, clinical relapse. Fig. 1C : Major form : small but multifocal lesions and high fungal density, clinical relapse after multiple surgery. Fig. 1D : typical major form: multifocal severe and high density, clinical relapse)” Position this as the footer of the figure 1.
Line 200. In results section, the authors show the findings of the histopathological examination of 21 patients. Of the 31 included patients, the histopathological was performed in 21? How the diagnosis was done in the other 10 cases? By direct examination of skin smear? Please, clear this point.
Line 204 and 205- Replace “size” for “diameter”.
Lines 207 to 216- (Figure 2A : Classic type, acanthosis and hyperkeratotic epidermidis, important inflammatory reaction of the dermis composed of multinucleated giant cells and histiocytes with numerous round yeasts, mostly intracellular, with some extracellular yeasts. A milder inflammatory infiltrant is also seen in the center and periphery of the lesion consisting of few lymphocytes, plasma cells and rare eosinophils. Figure 2B : Telangectatic scare type, the dermis is particularly fibrous and looks like a scare that delimites numerous dilated vessels. The scare is punctuated by a moderate inflammatory infiltrate consisting of histiocytes, lymphocytes and plasma cells with exceptional multinucleated giant cells with few yeasts which are mostly intracellular). Position this as the footer of the figure 2.
Line 218. "It is necessary to indicate in the caption of Figure 2 the staining used (hematoxylin-eosin) and the level of magnification applied to the image."
Lines 222 to 225. (Figure 3 : Patters of Lobomycosis yeast arrangement. We found three patterns of yeast arrangement. Fig A : small chain pattern of < 5 yeasts (Gomori-Grocott stain 40X). Fig B : long chain pattern of > 5 yeasts (Gomori-Grocott stain 40X). Fig C : rosette pattern corresponding to a round struc ture that usually has >5 yeasts showed by red arrow, Gomori-Grocott stain 60X)). Position this as the footer of the figure 3 and check if the level of magnification is 40x or 400x.
Line 234. It is necessary to clarify the meaning of "95% CI, [0.1-0.86]; p = 0.0476." Kindly incorporate the examined values into the text, including the 'n' and percentage of cases based on fungal density (low, median/high) and the number of lesions (unifocal, multifocal). This will enhance the transparency of these results.
Discussion
Line 253: Replace “Brazilain” with “Brazilian“.
Author Response
Reviewer 1
Comments
The authors have presented a well-written manuscript of a meticulously conducted study that evaluated the epidemiological and clinical features of 31 patients with lobomycosis in French Guiana. The histopathological examination was subjected to a reanalysis, leading the authors to identify two distinct clinical-histological patterns. In light of this, they propose a novel classification for the disease comprising two forms: the major form characterized by high fungal density and/or multifocal lesions, and the minor form characterized by low fungal density and unifocal lesions. Nonetheless, the manuscript could be further enhanced through improvements and clarifications. Hence, I would like to recommend the following adjustments:
Major review
- The authors state that “All patients diagnosed with lobomycosis in French Guiana between 1959 and 2022 were included”. How can the authors be certain that no other cases of lobomycosis existed in French Guiana during that time? What methodology was employed for the case search? Was an exhaustive search conducted across all healthcare facilities, as well as pathology and microbiology laboratories? Addressing this point is crucial, as the absence of cases was interpreted as a lack of occurrences. Could it be plausible that restricted access to the study hospitals during this period might have influenced the apparent absence of cases?
- We thank the reviewer for this comment. As the Cayenne Hospital Center is the only referral center for skin diseases and works closely with all other health care providers, all cases of Lobomycosis or other subcutaneous mycoses are systematically referred to its Dermatology Department. Suspicions of Lobomycosis in French Guiana are always confirmed either by the Mycology-Parasitology Laboratory or the Histopathology Laboratory of the Cayenne Hospital Center. Indeed, an exhaustive search was conducted in the databases of the Cayenne Hospital (Dermatology Department and the corresponding laboratories).
Therefore, we can assume that any case of lobomycosis that was diagnosed in French Guiana during the study period was included in our study. However, it is possible, due to barriers to health care access, that patients were not diagnosed during the study period.
- According Talhari & Talhari (2012) https://doi.org/10.1016/j.clindermatol.2011.09.014 , the diagnosis of lobomycosis can be effectively conducted through a direct examination of skin smears. This method is not only more accessible than histopathological examination, but also easier and more cost-effective. The authors of the study present a novel classification system that relies on histopathological findings. However, it's worth noting that histopathological examinations require the expertise of highly skilled medical professionals. Unfortunately, access to such professionals might be limited, especially in areas where the disease is endemic. In line with the authors' proposed algorithm, the clinical management of lobomycosis depends on the fungal density as determined through histopathological examination. Could the requirement for a histopathological examination potentially result in delays in diagnosis and treatment? Please, include a discussion about this topic in the manuscript."
We agree with the reviewer that histopathology results are not available before at least a few days, but it does not imply any treatment delay, as management can begin before receiving the histopathology results. The following sentences were added to the discussion “Diagnosis of Lobomycosis by direct examination of skin smears (Figure 6) enables rapid management of patients, with treatment instituted within a few days of diagnosis. Although histological examination on skin biopsy takes longer to obtain, initial management can be corrected secondarily according to the classification: proposal of adjuvant treatment in the case of a single lesion that has been operated and showing high fungal density and a high risk of recurrence (major form). “
- Why did the authors refrain from conducting statistical analyses to evaluate the association between the variables under study and the diagnosis period (group 1 and group 2) (results, line 143, table 1)? This would further substantiate the claim that there were a greater number of Brazilian patients and gold digger in group 2.
- We thank the reviewer for this comment. We performed a univariate analysis to sthrenghten our claim. In univariate analysis, Group 1 was associated with a Creole origin (OR 17.14, [95% CI 1.59-809.71] ; p=0.0039) while the occurrence of Brazilian patients (OR 0.02, 95% CI [0.01-0.21]; p=<0.0001) and gold mining (OR 0.19, 95%CI [0.03-1.19]; p=0.0384) as an occupation were significantly rarer than in Group 2.
These results are now available in Table 1 and in the results section.
Minor considerations:
Abstract
Line 23. “Conclusions: A resurgence of gold mining by Brazilian migrants in the 2000s changed the epidemiology of lobomycosis in French Guiana.” This statement presents an assumption that goes beyond the results obtained, a practice that should be avoid in the conclusion. Note that the same problem is on line 342 (conclusion section)
- We agree with the reviewer that we cannot formally assume that the resurgence of gold mining changed the epidemiology. However, the shift in nationality and occupation is striking. Therefore, we modified these sentences as follows “the changing epidemiology of lobomycosis in French Guiana was characterized by a shift towards Brazilian patients, mostly gold miners.”
Line 25. Please, note that the acronym MDT has not been spelled out before.
- MDTs stands for “multiple drug therapy”. It has been corrected as requested.
Methods
Line 107. Please, note that the acronym HE has not been spelled out before.
- HE stands for “hematoxyllin and eosin”. It has been corrected as requested.
Line 126. What test was performed to analyses “the association between the presence of a low fungal density and the risk of multifocal or unifocal presentations”? In the “results section”, line 234 it is unclear. (95% CI, [0.1-0.86]; p = 0.0476).
Line 130. “The relationship between individual risk factors and Lobomycosis infection was analyzed using Fisher’s exact test or χ2 test.”. There is a mistake here. The analysis did not involve assessing the associations between individual risk factors and Lobomycosis infection.
- We thank the reviewer for these two comments which are actually linked. This paragraph has been rewritten. Indeed, we did not look for associations between risk factors and Lobomycosis infection but the type of presentations and the fungal density, using Fisher or χ2 test (according to the number of entries for each variable.)
Results
Lines 166 to 168 “(Fig. 1A : Minor form. Unifocal lesion with low fungal density, complete response. Fig. 1B : Major form. unifocal but medium density, clinical relapse. Fig. 1C : Major form : small but multifocal lesions and high fungal density, clinical relapse after multiple surgery. Fig. 1D : typical major form: multifocal severe and high density, clinical relapse)” Position this as the footer of the figure 1.
- We thank the reviewer for this suggestion. All legends have been removed from the text and inserted as footers.
Line 200. In results section, the authors show the findings of the histopathological examination of 21 patients. Of the 31 included patients, the histopathological was performed in 21? How the diagnosis was done in the other 10 cases? By direct examination of skin smear? Please, clear this point.
- Actually, some patients had several samples. 21 samples corresponding to 13 patients were available. The following sentences were added: “A total of 21 samples (blocks and slides) of Lobomycosis were available in the archives of the Pathology Department, corresponding to 13 different patients. . Eight histological blocks corresponded to follow-up samples. Histological patterns and fungal densities were analyzed in the 21 available samples….As fungal density did not vary between initial and follow-up samples, we used the initial fungal density for statistical analysis (13 samples).”
- Concerning the diagnosis, all patients were confirmed either by histopathology or direct examination of skin smear. However, old histopathology samples (> 30 years) were not stored at the time of this study and could not be retrieved for analysis.
Line 204 and 205- Replace “size” for “diameter”.
- Theses changes have been made.
Lines 207 to 216- (Figure 2A : Classic type, acanthosis and hyperkeratotic epidermidis, important inflammatory reaction of the dermis composed of multinucleated giant cells and histiocytes with numerous round yeasts, mostly intracellular, with some extracellular yeasts. A milder inflammatory infiltrant is also seen in the center and periphery of the lesion consisting of few lymphocytes, plasma cells and rare eosinophils. Figure 2B : Telangectatic scare type, the dermis is particularly fibrous and looks like a scare that delimites numerous dilated vessels. The scare is punctuated by a moderate inflammatory infiltrate consisting of histiocytes, lymphocytes and plasma cells with exceptional multinucleated giant cells with few yeasts which are mostly intracellular). Position this as the footer of the figure 2.
Line 218. "It is necessary to indicate in the caption of Figure 2 the staining used (hematoxylin-eosin) and the level of magnification applied to the image."
- The legend of Figure 2 has been moved and modified according to comments from both reviewers.
Lines 222 to 225. (Figure 3 : Patters of Lobomycosis yeast arrangement. We found three patterns of yeast arrangement. Fig A : small chain pattern of < 5 yeasts (Gomori-Grocott stain 40X). Fig B : long chain pattern of > 5 yeasts (Gomori-Grocott stain 40X). Fig C : rosette pattern corresponding to a round struc ture that usually has >5 yeasts showed by red arrow, Gomori-Grocott stain 60X)). Position this as the footer of the figure 3 and check if the level of magnification is 40x or 400x.
- This legend has been moved accordingly.
Line 234. It is necessary to clarify the meaning of "95% CI, [0.1-0.86]; p = 0.0476." Kindly incorporate the examined values into the text, including the 'n' and percentage of cases based on fungal density (low, median/high) and the number of lesions (unifocal, multifocal). This will enhance the transparency of these results.
- We added the corresponding values. Please note that by suing Fisher’s exact test, the p-value has slightly changed. “5 patients had a low fungal density, all of them had unifocal presentation. Among 8 patients with a medium or high density, 5 had a plurifocal presentation and 3 a unifocal one. A low fungal density was associated with a lower risk of a multifocal involvement (OR 0.1 ;95% CI, [0.01-0.68]; p = 0.044).”
Discussion
Line 253: Replace “Brazilain” with “Brazilian“.
- Modified accordingly.
Although a more thorough study would include studying active patients and performing anti-fungal susceptibilities, the goal of this study was mainly historically descriptive, perhaps with a purpose to “freshen” the literature base for this rare fungal disease.
Discussion should include the limitation that fungal arrangement and density in tissues may be related to immune responses which weren’t measured. Another limitation was that antibodies and cellular immunity were not measured. Is it possible that Th2 immunity predisposes to multifocal infection and poor outcomes and chronicity?
- We agree with the reviewer on the useful of immune studies, which we were not able to perform at the time of study, due to technical constraints. The following sentences were added to the discussion: “Another limitation is that immunohistochemistry was not available for this study. Fungal density and histopathological patterns may be related to immune responses. Further studies should look for the role of host factors through measures of antibodies and cellular immunity.” It is likely indeed that Th2 immunity could play a role in the severity of lobomycosis, as it is known that a dominance of T reg activity inhibits the immune response.
- Concerning antifungal susceptibilities, to the best of our knowledge, it is very difficult to perform such tests on lobomycosis, as this fungus does not grow in mycological culture. Of course this information would be very useful to understand the different therapeutic responses.
Minor:
Line 213 shoud be “scar”, not scare.
- Modified accordingly.
Figure 3 figure legend appears to be in the wrong place. Arrows in the figure would be beneficial to highlight the features described in the text.
- Figure 3 legend has been moved as mentioned to Reviewer 1. Other arrows were added for further clarity.
Reviewer 2 Report
Grotta et al. describe the clinical and therapeutic features of lobomycosis in French Guiana from 1959 to 2002. They also describe histopathologic features of lobomycosis cases and propose a prognostic clincio-histologic classification of patients to guide therapeutic treatment.
Although lobomycosis is rare it is important to track the epidemiological and geographic shifts that have occurred over the past 50 years. It is also important to document gross skin lesions as well as microscopic histologic findings from tissue sections to guide diagnosis and treatment. Finally, outcomes in response to various anti-fungal treatments are important to document so that subsequent cases may benefit from historical findings.
The introduction would benefit from a statement of where the fungus is found in nature and how it reproduces independent of human hosts.
Skin smear image of a lesion stained with Gomori-Grocott would also be helpful to readers.
Figure 2 would benefit from higher power microscopy of the features described in the text including multi-nucleated giant cells with a better description in the figure legend of what the reader should observe in the figure.
The study would also benefit from immunohistochemistry for B cells, T cells and macrophage markers. IHC for cytokines would be welcome, but not necessary here.
Although a more thorough study would include studying active patients and performing anti-fungal susceptibilities, the goal of this study was mainly historically descriptive, perhaps with a purpose to “freshen” the literature base for this rare fungal disease.
Discussion should include the limitation that fungal arrangement and density in tissues may be related to immune responses which weren’t measured. Another limitation was that antibodies and cellular immunity were not measured. Is it possible that Th2 immunity predisposes to multifocal infection and poor outcomes and chronicity?
Minor:
Line 213 shoud be “scar”, not scare.
Figure 3 figure legend appears to be in the wrong place. Arrows in the figure would be beneficial to highlight the features described in the text.
Author Response
Comments
Grotta et al. describe the clinical and therapeutic features of lobomycosis in French Guiana from 1959 to 2002. They also describe histopathologic features of lobomycosis cases and propose a prognostic clincio-histologic classification of patients to guide therapeutic treatment.
Although lobomycosis is rare it is important to track the epidemiological and geographic shifts that have occurred over the past 50 years. It is also important to document gross skin lesions as well as microscopic histologic findings from tissue sections to guide diagnosis and treatment. Finally, outcomes in response to various anti-fungal treatments are important to document so that subsequent cases may benefit from historical findings.
The introduction would benefit from a statement of where the fungus is found in nature and how it reproduces independent of human hosts.
- We thank the reviewer for this suggestion. The following sentence was added to the introduction: “The main hypothesis is that the reservoir of the fungal agent is found in tropical waters and on trees in the Amazon rainforest, and could also contaminate animals and tools. Indeed, a history of trauma is often found years before the appearance of lobomycosis lesions. It has been reported that byit is described that skin lesions can occured after following animal bites (such as fish or insects bites) or cuts by plants or sharp objects would lead to inoculation”.
Skin smear image of a lesion stained with Gomori-Grocott would also be helpful to readers.
- We thank the reviewer for this suggestion. An image of skin smear has been added as Figure 6.
Figure 2 would benefit from higher power microscopy of the features described in the text including multi-nucleated giant cells with a better description in the figure legend of what the reader should observe in the figure.
- We thank the reviewer for this suggestion. Figure 2 has been modified accordingly.
The study would also benefit from immunohistochemistry for B cells, T cells and macrophage markers. IHC for cytokines would be welcome, but not necessary here.
- We agree with the reviewer that immunohistochemistry could yield very useful results. However, these blocks have since been deparaffined and used for genotyping tests. Therefore, the material is not available anymore and we will not be able to perform any IHC study.
Round 2
Reviewer 1 Report
The authors have addressed all requests, and I support accepting the manuscript for publication in its current state.
Reviewer 2 Report
Accept with minor changes submitted to JoF editors.
Line 124: replace “destored” with “retrieved.”
Line 159: Instead of “Associations were looked for between…” It should read: “We also looked for associations between…”
Needs a copy-editor to make sure English is clear.